# EMPIRICAL SUFFICIENCY FEATURING REWARD DELAY CALIBRATION

## ABSTRACT

Appropriate credit assignment for delay rewards is a fundamental challenge in various deep reinforcement learning tasks. To tackle this problem, we introduce a delay reward calibration paradigm inspired from a classification perspective. We hypothesize that when an agent's behavior satisfies an equivalent sufficient condition to be awarded, well-represented state vectors should share similarities. To this end, we define an empirical sufficient distribution, where the state vectors within the distribution will lead agents to environmental reward signals in consequent steps. Therefore, an overfitting classifier is established to handle the distribution and generate calibrated rewards. We examine the correctness of sufficient state extraction by tracking the real-time extraction and building hybrid different reward functions in environments with different levels of awarding latency. The results demonstrate that the classifier could generate timely and accurate calibrated rewards, and the rewards could make the training more efficient. Finally, we find that the sufficient states extracted by our model resonate with observations of human cognition.

## 1 INTRODUCTION

Reinforcement learning (RL) approaches have made incredible breakthroughs in various domains (Silver et al., 2016; Mnih et al., 2015; OpenAI et al., 2019; Vinyals et al., 2019), where the performance exceeds people's expectations. The reinforcement learning theoretical basis models sequential decision tasks as dynamic programming processes to maximize expected accumulated rewards. Given that environmental rewards generally cannot entirely reflect the contribution of each action in a step, existing approaches commit to distributing different credits to individual decisions, known as credit assignment (Sutton & Barto, 1998). Bellman equation-based architecture calculates a value of a state based on the gathered rewards in the future, which at times assigns an unreasonable value to prior states. This problem becomes even more intractable when reward signals are extremely sparse or severely delayed.

In this paper, we formulate an overfitting classification mechanism to extract empirical sufficient conditions of acquiring desired environmental signals. We refer to this extraction formulation as an Empirical Sufficient Condition Extractor (ESCE) to fairly assign delayed rewards to corresponding states. In so doing, we first propose a classification mechanism to identify empirical sufficient states. To train a classifier with partially labeled data, we label the state vectors with matched environmental signals. We then train the classifier with two phases, wherein a novel overfitting training process is conducted. In addition to existing value-based estimation, the ESCE provides concrete predictions. We equip the ESCE with Asynchronous Advantage Actor Critic (A3C) algorithm (Mnih et al., 2016) and measure the performance on six Atari games, most of which have delayed discrete rewards. We comprehensively examine the extraction correctness by formulating different reward functions, and further track the accuracy/recall of ESCE on the fly. The results show that agents guided by our empirical efficiency achieve significant improvements in convergence, especially in the scenarios with delayed rewards. Furthermore, we constructively modify the environment to render the rewards even to be more delayed, termed as *hindsight rewards* settings. The results show that equal calibrated rewards could lead agents to acquire well-learned target policies as if rewards are not delayed. In addition to quantitative experiments, we screenshot the identified sufficient states, showing high similarity with human cognition. Our contributions can be summarized as follows:

- We introduce an overfitting classification model to extract empirical sufficient conditions, where overfitting mechanism could significantly reduce uncertainty.

- We formulate a calibrated reward signal in line with the environmental targets to tackle the reinforcement learning reward delay issues, such that the rewards are provided when empirical sufficient conditions are satisfied.

- The experimental results show reward-calibrated agents are able to learn decent target policies in the scenarios where rewards have been severely delayed. The identified sufficient conditions empirically resonate with the "true" environment targets.

## 2 RELATED WORK

### 2.1 INTRINSIC MOTIVATION

Intrinsic rewards (Singh et al., 2004; Ryan & Deci, 2000), inspired by intrinsic motivation, are primarily introduced to encourage exploration. The rewards mechanism is largely independent of environmental rewards. Intrinsic Rewards in the exploration-oriented mechanism are generally correlated to the novelty or informative acquisition of new arrival states (Pathak et al., 2017; Burda et al., 2019; Houthooft et al., 2016; Zhang et al., 2019). Due to the awarding mechanism does not depend on environmental rewards, as an exchange, the policy may not align with the environmental target. In addition to exploration, intrinsic rewards can often be found in hierarchical frameworks (Kulkarni et al., 2016; Vezhnevets et al., 2017; Frans et al., 2018). Moreover, intrinsic rewards are also used to assist agents to more directly learn optimal or near-optimal policies (Wang et al., 2020; Zheng et al., 2018; 2019). Likewise, following down this branch, the ESCE developed in this paper could generate empirical intrinsic rewards to learn better policies, without encouraging exploration.

### 2.2 CREDIT ASSIGNMENT FOR DELAYED REWARDS

Most evaluation mechanisms in reinforcement learning rely on Bellman equation, where the environmental signals are passed across states in sequences (Lee et al., 2019; Arjona-Medina et al., 2019; Ng et al., 1999; Marom & Rosman, 2018). To make the training more efficient, one effective direction is to build an extra mechanism to capture critical states and to emphatically regress on these states (Sutton et al., 2016; Ke et al., 2018; Hung et al., 2018). Ideologically, Irpan et al. (2019) introduce binary classification into evaluation, and positive-unlabeled learning (Kiryo et al., 2017) is adopted to distinguish promising and catastrophic states. Our work evaluates states by discriminating states as a simple binary classification problem without relying on Bellman equation. Specially, we accurately differentiate states between "sufficient for success" and "insufficient for success", by developing a new overfitting classifier (see Sections 3.3 and 3.4).

## 3 REWARD DELAY CALIBRATION

### 3.1 LEARNING WITH HYBRID REWARD FUNCTIONS

As an independent module, the proposed Empirical Sufficient Condition Extractor (ESCE) can be incorporated into multiple mainstream reinforcement learning frameworks. Calibrated rewards are provided by ESCE when a state meets the empirical sufficient condition. We denote $\pi(s_t; \theta_P)$ as the learned policy, where $\theta_P$ is the set of parameters of the policy network; $r_t^c$ is the calibrated reward generated by ESCE and $r_t^e$ an environmental reward from the environment, at time step $t$. Both reward signals have the same scale. The total reward function is synthesized with calibrated signals and environmental signals, $r_t = \alpha r_t^c + \beta r_t^e$, where $\alpha$ and $\beta$ are the weight coefficients of corresponding rewards. Our baseline, optimized by environmental rewards, has coefficients $\alpha = 0$ and $\beta = 1$. The policy network is optimized to maximize expected accumulated rewards, as shown below:

$$\max_{\theta_P} \ \mathbb{E}_{\pi(s_t; \theta_P)} \Sigma_t r_t. \tag{1}$$

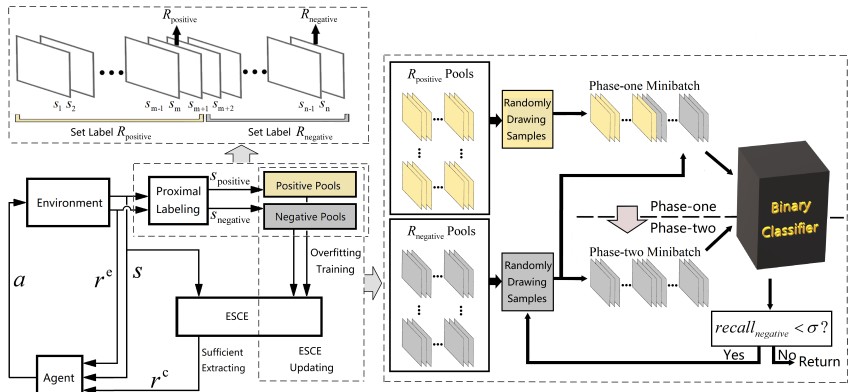

Figure 1: The overall framework of a RL agent equipped with ESCE. Agents receive environmental rewards and calibrated signals from ESCE as its total rewards. The policy network and ESCE are two independent models. The ESCE examines each state and provides calibrated rewards when the state is considered as an empirical sufficient state. Within the ESCE training process, state vectors are automatically labeled with our proposed novel labeling method, and are then stored in corresponding pools. The ESCE network is updated with overfitting training, where phase one is a binary classification with data from pools and $R_{\text{negative}}$ pools, and phase two is the proposed overfitting training updated with $R_{\text{negative}}$ data alone.

The overview of our framework is illustrated in Figure 1. Whenever an empirical sufficient state is identified by ESCE, a positive reward is offered. The calibrated reward is available after receiving an environmental signal. Meanwhile, these states are labeled and stored in pools. The training of the ESCE and policy network proceeds alternately. We take as input raw image pixels without any data preprocessing.

## 3.2 EMPIRICAL SUFFICIENT DISTRIBUTION (ESD)

The occurrence of a particular event's sufficient condition is always followed by the occurrence of its corresponding event. Inspired by this, we try to resolve the reward delay issue in reinforcement learning from a similar perspective. We consider a set of particular environmental signals as target events, and utilize the information from state feature vectors to yield the sufficiency to incur these target events. We hypothesize some vectors are closely distributed to each other, if: (1) the state vectors contain all critical information, and (2) the state vectors include the information of a same sufficient condition. We therefore use this hypothesis to proceed with the classification-based evaluation.

**Definition 3.1. Empirical Sufficiency.** We define empirical sufficiency if there exists a bounded space such that all reachable state vectors inside always imply a particular environmental reward signal.

Given a stable environment, let $R_{\text{positive}}$ be desired environmental signals, such as positive rewards. Conversely, we use $R_{\text{negative}}$ to represent undesired environmental signals, including negative rewards, agent's deaths, game endings, etc.

We define Policy-based Empirical Sufficient Distribution (ESD-*policy*) as: Agents explore the environment with a specific policy; if an agent reaches a state $s_{\text{suf}}^{\pi}$, invariably acquiring $R_{\text{positive}}$, we take a state $s_{\text{suf}}^{\pi}$ as a policy-based empirical sufficient state to acquire $R_{\text{positive}}$. If all reachable states in a bounded distribution are empirical sufficient states, then we consider the distribution as a Policy-based Empirical Sufficient Distribution of $R_{\text{positive}}$. In this case, the ESD-*policy* is effective if: (1) the environment remains stable, and (2) the policy remains stable. If the policy is upgraded, the corresponding ESD-*policy* may change as well.

## 3.3 LABELING

We implement ESCE training with a binary classification mechanism and take individual raw pixels of states as input. For labeling, we automate this process to extend environmental signals as labels for classification training.

We first cut a long sequence of states into rounds along with environmental signals, and use these labels as break points of rounds. Furthermore, these environmental signals are adopted as the labels for states within a round. Specifically, we use a newly received environmental signal as a mutual label for states starting from the last environmental signal to the newly received signal. The labeling procedure is shown in Figure 1. We generalize all desired environmental signals as $R_{\text{positive}}$, and all undesired environmental signals as $R_{\text{negative}}$. Correspondingly, $R_{\text{positive}}$ and $R_{\text{negative}}$ pools are created for collecting labeled pixels.

As we adopt nearby environmental signals as labels, another problem is label ambiguity. For the states whose endings are not determined yet, the labels can be converse in different episodes, such as beginning states. To put it differently, the state vectors of two different labels might be densely mixed, and there might not be a clear boundary between them, as opposed to supervised training, where every sample has only one corresponding label. Our classification training procedure is inspired by the label ambiguity. Regarding Definition 3.1, ESD should only contain states that lead to one particular environmental signal. This indicates that ESD should only include states with unique labels. We formulate ESCE training as an overfitting training process to exclude state space with ambiguous labels.

## 3.4 OVERFITTING TRAINING

We formulate a two-phase training process to extract pure distribution. In phase one, we expect ESCE to assign a dominant probability to real empirical sufficient states since ambiguous states may fool the classifier. In phase two, to exclude insufficient states, an overfitting training mechanism is adopted to update the decision boundary.

We measure performance and maturity of ESCE with *Precision* and *Recall* on $R_{\text{positive}}$. The *Recall* roughly indicates the ratio of identification coverage on those samples associated with $R_{\text{positive}}$, and the *Precision* roughly shows how accurate the identification is. We set the number of rounds that include identified states as $N_{\text{ident}}$, and let $N_{\text{sufficient}}$ denote the number of rounds that contains identified empirical sufficient states and leads to $R_{\text{positive}}$. Additionally, we set the total number of samples leading to $R_{\text{positive}}$ acquired as $N_{\text{pos}}$.

The phase one is carried out by a binary classification training. The training data are from both $R_{\text{positive}}$ and $R_{\text{negative}}$ pools, where all $R_{\text{positive}}$ samples ($s_{\text{pos}}$) have desired signal labels ($r_{\text{pos}}$), and $R_{\text{negative}}$ samples ($s_{\text{neg}}$) have undesired signal labels ($r_{\text{neg}}$). We use both types of samples to prudently maximize *Recall* on the basis that ESCE parameters $\theta_{\text{E}}$ are trained with an appreciable amount of representative samples (Figure 2(a)). Let the learning function $f$ generate estimates of foreseeable reward $\hat{r}$ with ESCE network parameters $\theta_{\text{E}}$, where $\hat{r}_{\text{pos}}$ and $\hat{r}_{\text{neg}}$ are respectively generated with $s_{\text{pos}}$ and $s_{\text{neg}}$, as defined below:

$$\hat{r}_{\text{pos}} = f\Big(s_{\text{pos}}; \theta_{\text{E}}\Big), \tag{2}$$

$$\hat{r}_{\text{neg}} = f\Big(s_{\text{neg}}; \theta_{\text{E}}\Big). \tag{3}$$

To reflect the real network identification of samples, the output is returned by the maximum likelihood estimate of the soft-max distribution rather than sampling. The loss of phase one measures the discrepancy between estimates and foreseeable signals, which are defined as follows:

$$\begin{aligned} \max_{\theta_{\text{E}}} \ (\text{Precision}_{\text{positive}}) &= \max_{\theta_{\text{E}}} \ \left(\frac{N_{\text{sufficient}}}{N_{\text{ident}}}\right) \\ &\approx \min_{\theta_{\text{E}}} \ L_E^{\text{one}}\Big((\hat{r}_{\text{pos}}, r_{\text{pos}}) + (\hat{r}_{\text{neg}}, r_{\text{neg}})\Big). \end{aligned} \tag{4}$$

In phase two, we maximize *Precision* with complete $R_{\text{negative}}$ samples. The classification boundary is updated to acquire a distribution of pure $R_{\text{positive}}$ samples by excluding all insufficient samples. The signal that the ESCE has sufficiently excluded insufficient distribution appears when the *Recall* of $R_{\text{negative}}$ sample batch reaches 100% ($\sigma$) (see Figure 2(b)). The loss of phase-two process is defined as follows:

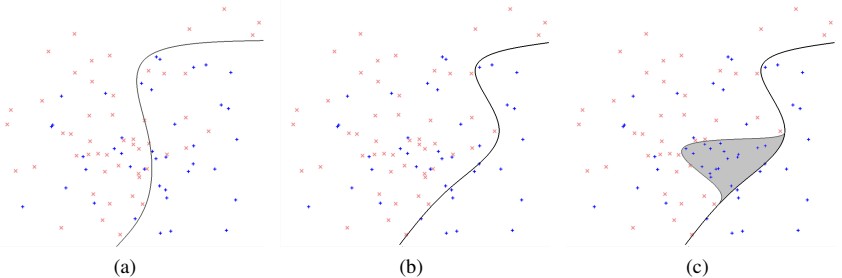

(a)  (b)  (c)

Figure 2: 2(a) A schematic diagram of decision boundary after phase-one overfitting training: the classifier will try to make most of the samples correctly distributed on both sides of the boundary. However, there are still quite a few promiscuous samples falsely allocated due to label ambiguity. 2(b) A schematic diagram of decision boundary after phase-two overfitting training: all state vectors labeled with $R_{\mathrm{negative}}$ are excluded from $R_{\mathrm{negative}}$ boundary, which matches the definition of ESD-*policy*. 2(c) A schematic diagram of sensitive Sampling: the actual ESD-*policy* may change with the policy's updating. The state vectors inside the gray region were insufficient states, and then turn into empirical sufficient states. The ESCE is primarily updated with these dynamic samples.

$$\max_{\theta_{\mathrm{E}}} \ (\mathrm{Recall}_{\mathrm{positive}}) = \max_{\theta_{\mathrm{E}}} \ \left( \frac{N_{\mathrm{sufficient}}}{N_{\mathrm{pos}}} \right)$$
$$\approx \min_{\theta_{\mathrm{E}}} \ \sigma L_E^{\mathrm{two}} \left( \hat{r}_{\mathrm{neg}}, r_{\mathrm{neg}} \right). \tag{5}$$

Once phase-two process is terminated, the states recognized as $R_{\mathrm{positive}}$ should only include states with label $R_{\mathrm{positive}}$. In other words, all states identified as $r_{\mathrm{pos}}$ should have resulted in $R_{\mathrm{positive}}$, in line with the definition of empirical sufficient distributions. If the network is updated with the policy and the sampling is representative, the distribution identified would be the Policy-based Empirical Sufficient Distribution (ESD-*policy*). Theoretically, if stochastic policies are adopted without any bias, Task-Specific Empirical Sufficient Distribution (ESD-*task*) may be acquired. We show the ESCE architecture in Algorithm 1, which can be found in the Appendix.

The optimization for overall architecture could be derived through combining Equations (1), (4), and (5):

$$\min_{\theta_{\mathrm{P}}, \theta_{\mathrm{E}}} \ \left( - \mathbb{E}_{\pi(s_t; \theta_P)} \Sigma_t r_t + L_E^{\mathrm{one}} + \sigma L_E^{\mathrm{two}} \right). \tag{6}$$

### 3.5 Sensitive Sampling

Given ESD-*policy* is based on a particular policy defined in Section 3.2, ESD-*policy* might change with corresponding policy's updating. To efficiently update the ESCE network, we adopt a sensitive sampling strategy for overfitting training.

For ESD-*policy*, the empirical sufficient distribution of $R_{\mathrm{positive}}$ should change as the agent is updated. For instance, if an agent becomes more skillful, it may stably acquire rewards which were previously unattainable. As a result, the physical volume of ESD-*policy* would inflate. An efficient way to update the ESCE parameter $\theta_E$ is to focus on the evolving data distribution (see Figure 2(c)). Accordingly, we build two sensitive state pools. One pool is for missed identification, indicating which rounds reach $R_{\mathrm{positive}}$, without identified states in the sequences; the other pool is built for false identification, which collects the rounds containing identified pixels that did not actually reach $R_{\mathrm{positive}}$. These two state pools force ESCE to focus on the variation of policy, either new strategy learned or strategy forgotten due to the network update. Approximately 75% of training data are imported from two sensitive pools.

Table 1: Comparison with baselines on hindsight rewards settings: $r_t^c = 1 \cdot r_t^c + 0 \cdot r_t^e$. Rewards are offered after a $R_{\text{negative}}$ signal or the end of episodes; the real time *Precision* and *Recall* of $R_{\text{positive}}$ are synchronized with time steps, which are referred to as the right vertical axis.

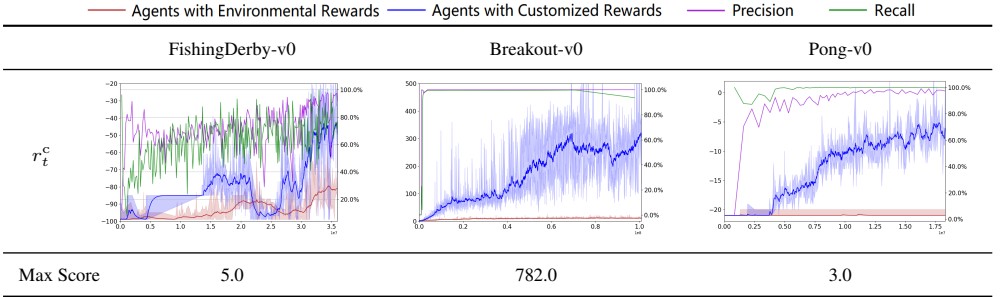

| | FishingDerby-v0 | Breakout-v0 | Pong-v0 |
|---|---|---|---|
| Max Score | 5.0 | 782.0 | 3.0 |

## 4  EXPERIMENTAL RESULTS

We primarily measure the ESCE component by building different reward functions, and compare to the baseline A3C+LSTM agents on six Atari games. The original A3C LSTM agent only optimizes with environmental rewards, written as $r_t = 0 \cdot r_t^c + 1 \cdot r_t^e$. The detailed experiment settings are laid out in Appendices A.1 and A.2. Further, we examine agents' performance on modified experimental settings, termed as *hindsight rewards* settings, where the summation of environmental rewards is provided after a short episode.

Since it is unusual and risky to schedule overfitting as a part of the training, the intuition behind the design is that the spatial expansion of the $R_{\text{negative}}$ identification potentially makes the space of $R_{\text{positive}}$ inevitably shrink. It is worth worrying how overfitting affects the recognition on $R_{\text{positive}}$ samples. We therefore try to raise two research questions in our experiments:

- RQ1: Does the severely overfitting network perform well, and what are the results of the extraction?

- RQ2: How do calibrated rewards affect RL training?

We attempt to answer RQ1 in Sections 4.1 and 4.2 and answer RQ2 in Sections 4.3 and 4.4.

### 4.1  PRECISION AND RECALL OF EMPIRICAL SUFFICIENT STATE TRAINING

To answer RQ1, we first examine the training process of empirical sufficient extraction statistically. The *Recall* indicates how widespread the identification is, and the *Precision* indicates how accurate the prediction is.

It is rare, although almost unlikely, that the boundary trained perfectly matches the "true" boundary of the sufficient distribution. This means the *Recall* and *Precision* of positive samples are the pair of trade-offs. Empirically, we fine-turned the hyper-parameter $\sigma$ from 0.81 to 1, which in turn keeps *Recall* and *Precision* both high. The changes of these two indices are recorded in Tables 1 and 2, in which the right vertical axes correspond to the two indices. We also identify that *Precision* has a strong positive correlation with the policy. Since the policy networks are randomly initialized at the beginning, the erratic performance makes the prediction difficult. **In most games, both *Recall* and *Precision* could reach high values when the sampling policy becomes stable**.

For Breakout-v0 shown in Table 2, the *Recall* significantly reduces when agents on average get more than 40 marks in the game as the bricks hit by the pellet could be vastly different in every episode. Thus, when compared with other games, there are more state variance in Breakout-v0 environment. Given that the overfitting mechanism is adopted in our model, its ability to eliminate negative samples differentiates across models. Therefore, when compared to other networks that do not account for overfitting, the ESCE model is more sensitive to state change.

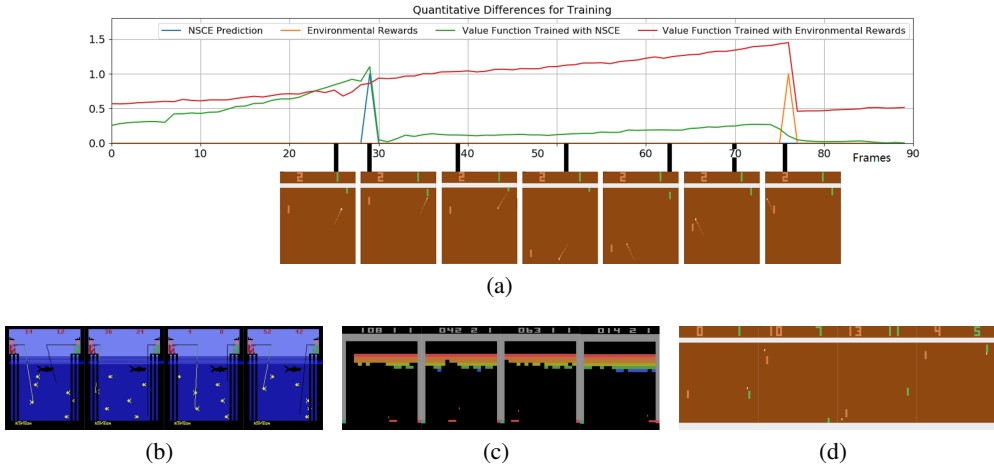

(a)

(b)                         (c)                         (d)

Figure 3: 3(a) The calibrated rewards and environmental rewards are encoded in red and yellow, and the value estimation (Critic) trained with calibrated rewards and environmental rewards are encoded in green and red, respectively. The blue line displays that ESCE identifies a empirical sufficient state when the agent hit the pellet with the edge of the bat (right bat), which significantly increases transverse velocity for the agent to win the game, and makes it unsolvable for its opponent. However, the value of baseline (red) increases continuously until receiving the reward returned by the environment, which is far away from the decisive state. To the best of our knowledge, prior approaches could not make such precise prediction (blue) on critical states. 3(b) In FishingDerby-v0, most states are identified when the agents' hook close to fishes or a fish is already hooked. 3(c) For Breakout-v0, most recognition occurs when the pellet is close to the bat or the bat is on the pellet's potential trajectory. 3(d) For Pong-v0, the recognized states show that the opponent is about to miss, or agents hit the pellet with the edge of the bat, which gives pellets has a quick vertical velocity.

## 4.2 EMPIRICAL EFFICIENCY IDENTIFICATION

To further verify the correctness of ESCE, we screenshot the extracted states on three games. **Most states identified exhibit high correlations with rewards acquisition and are visually close to human cognition**. With the evolving of the agent policy, ESD-*policy* keeps altering, which results in the dynamically changed observations. In the initial episode, the empirical sufficient checkpoints are recognized only a few steps away from the states where the actual rewards are given; with the policy becoming stronger and stabler, more and earlier states can be identified. We thus infer that a well-trained ESCE model is capable of making accurate predictions through the deep understanding of the policy and environment. We visualize the screenshots of extracted states in Figure 3.

## 4.3 REWARD DELAY CALIBRATION IN HINDSIGHT REWARDS SETTING

The latency in realistic environments can be highly unpredictable. Since current experiments are still largely unable to reflect the performance of different models on reward delay environments, we further modify the environments to force it to offer more delayed rewards, which we termed as hindsight rewards. In this setting, rewards are only provided if an episode ends or after a negative environment signal. We compare the performance of two agents trained with environment rewards and calibrated rewards in the hindsight rewards settings, respectively. The results of the three games are shown in Table 1. In this modified scenario, agents guided by environment rewards can hardly make any progress, whereas **the ones updated with calibrated rewards are able to learn distinctive target policies**.

The reason behind this phenomenon is that the ESCE model helps to calibrate the reward by identifying those states that meet the empirical sufficient condition. Given that the value is gained from the acquired rewards, rewards received from inappropriate states would thus mislead the value estimation. Figure 3(a) illustrates the difference of agent value estimation learned with environmental rewards and calibrated rewards. As seen from Figure 3(a), the value function (Critic) trained with environmental rewards and calibrated rewards are encoded in red and green, respectively. The calibrated reward (blue) is offered beforehand compared with the environment reward (yellow). This

Table 2: Comparison between different reward functions and original environmental rewards. Row-one: $r_t = 0.3 \cdot r_t^c + 1 \cdot r_t^e$. Row-two: $r_t = 1 \cdot r_t^c + 0 \cdot r_t^e$. The real time *Precision* and *Recall* of $R_{\text{positive}}$ are synchronized with time steps, which are referred to as the right vertical axis. For Bowling-v0, the data size is too small to provide valid *Precision* and *Recall* statistics.

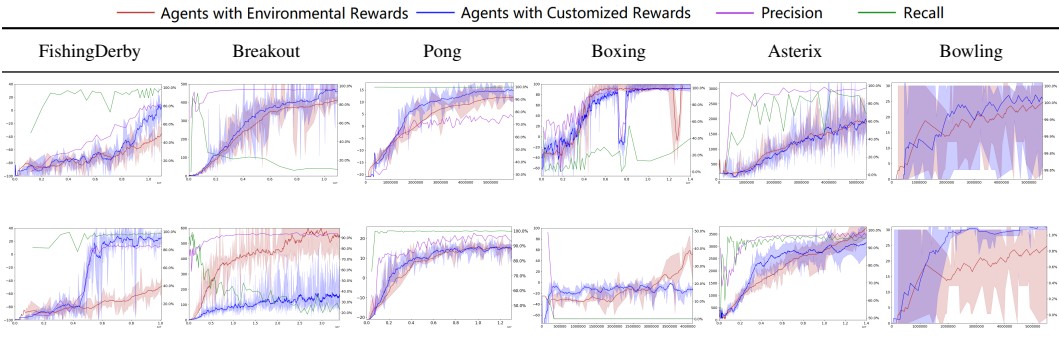

is because the ESCE identifies an empirical sufficient state at the moment when the agent hits the pellet with the edge of the bat.

## 4.4 FULLY- AND SEMI-CALIBRATED REWARDS

In addition to modified hindsight rewards settings, we also build two reward settings to explore how the calibrated rewards affect the RL training process. The first is the semi-calibrated rewards setting, where the calibrated reward coefficient is set to 0.3 and the environmental reward coefficient is 1, written altogether as $r_t = 0.3 \cdot r_t^c + 1 \cdot r_t^e$. The second is the fully-calibrated rewards setting, where the reward function is $r_t = 1 \cdot r_t^c + 0 \cdot r_t^e$. The results are shown in Table 2.

For the semi-calibrated rewards setting, the results show that **a small number of calibrated rewards could accelerate the training process**. It is also clear that higher *Recall* and *Precision* are essential prerequisites to ensure the effectiveness of calibrated rewards toward environmental targets. In Breakout-v0 and Boxing-v0, the *Recall* can barely reach a high value, and we believe this is due to the large variance in their states (see Section 4.2). Essentially, for Boxing-v0 the two players could be found in any position of the screen, while for Breakout-v0 the remaining bricks could be vastly different at each episode.

For the fully-calibrated rewards setting, only calibrated rewards are provided when reaching an empirical sufficient condition. This ablation comparison examines the rationality of the time of awarding. The results show that **agents trained with calibrated rewards can beat our baseline model in multiple different games.** To acquire rewards in FishingDerby-v0, agents need to move a hook to catch fish, and then reel back the line before a shark eats the fish. It is common that the shark steals the fish if the agents have not learned strategies to quickly pull the hook up. In the first column of Table 2, the calibrated rewards significantly boost the convergence of the model, causing a spike. This is because when the agent learns a stable policy of pulling the hook up, the empirical sufficient condition simply becomes hooking the fish. The ESCE provides instant calibrated rewards to the action triggering the state of successfully hooking the fish. This training process also resembles the human learning strategy by breaking down complex missions into easier sub-tasks.

## 5 CONCLUSION

In this paper, we formulate an approach to calibrating delay rewards from a classification perspective. Due to the overfitting mechanism, the proposed ESCE model is capable of accurately extracting the critical states. Accordingly, the agents trained with calibrated rewards could assign supreme values to the critical states. In addition, the results show that agents trained with calibrated rewards could learn distinctive target policies in environments with extremely delayed rewards. Furthermore, the screenshots of extracted states show a strong correlation with environmental rewards, visually approaching the observations of human cognition.

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

# A APPENDICES

## A.1 SPARSE AND DELAY REWARD SETTING

We evaluate our model and the comparing models on 6 Atari 2600 games on OpenAI Gym (Brockman et al., 2016), and uniformly choose v0 of all games, where actions are not repeated. Positive rewards are defined as $R_{\text{positive}}$, while the negative rewards, deaths, and game endings are $R_{\text{negative}}$. In some games, the environmental rewards are offered in a few steps after the empirical sufficient state. For instance, in the game Bowling-v0, the environmental rewards are determined when the character throws the ball. Reinforcement Learning algorithms are inevitably facing the low sample efficiency issue, while the delayed rewards worsen the problem. The longer gaps in time between empirical sufficient state and reward are, the more challenge the agent has to learn the "correct" policies.

## A.2 A3C AND ESCE ARCHITECTURE

We adopt an A3C + LSTM framework as our backbone architecture. The original RGB value $210\times160$ image frames are converted to $80\times80$ gray-scale frames. Four continuous frames are stacked as the input. In A3C architecture, four convolution layers and max-pooling layers are adopted. An LSTM layer with 512 units is followed with two heads — a policy head and a value function head.

Considering the policy is evolving on the fly, the policy-based empirical sufficient conditions should be updated accordingly. To ensure the extracted empirical sufficient distribution is up to date with the policy, we train the ESCE model with data sampled from the latest episodes. Subject to this reason, the maximum capacity of the datasets for the training of empirical sufficient state classifier should be flexibly decided. On the other hand, training models in model-free settings request vast samples for convergence. Thus, it is necessary to expand datasets to cover more cases. After trading-off between covering more cases and minimizing the variance of collected sample points, the capacity of $R_{\text{positive}}$ and $R_{\text{negative}}$ pools is set from 20,000 frames to 80,000 frames, sampled by 24 workers.

---

**Algorithm 1:** Empirical Sufficient Conditions Extractor (ESCE)

---

Initialize Extractor network and policy $\theta$;
Initialize $R_{\text{positive}}$ pool and $R_{\text{negative}}$ pool;
**repeat**
    Initialize temporary storage;
    **while** *any state pool is not full* **do**
        **if** *state $S_t$ is identified as $R_{\text{positive}}$ and no calibrated reward have been given to the*
        *agent in this round* **then**
            Assign calibrated reward to the agent;
        **end**
        Push states to a temporary storage;
        Update the policy network with the parameters $\theta$;
        **if** *environmental signal is not null* **then**
            Push temporary storage to pool based on positive/negative environmental signals;
            Clear temporary storage;
            Reset calibrated awarding status (Start a new round);
        **end**
    **end**
    Update Extractor with states from both pools;
    **while** *Recall of $R_{negative} < \sigma$* **do**
        Update ESCE parameters with data from $R_{\text{negative}}$ pool;
    **end**
    Clear all pools;
**until** *Converged*;

---

