# OpenReview forum: "Empirical Sufficiency Featuring Reward Delay Calibration"
_ICLR.cc/2021/Conference — Reject_

### Official Review · AnonReviewer3 · 2020-10-19
**The proposed method seems to be ad-hoc, and neither has a good theoretical explanation, nor there is any rigorous experiment to validate and compare against the existing methods.**

**Rating:** 4
**Confidence:** 3

**Review:**

The paper presents an experimental approach to tackle the problem of appropriate credit assignment under the delayed reward scenario and proposes an empirically sufficient condition to calibrate the reward policy in a Deep Reinforcement Learning framework.

**Pros**
    - The problem is interesting.

**Cons**
    - The paper does not explain the pre-processing in sufficient depth. In the experiments (neither in section 4 nor in A.1 or A.2) the state representation is not clear.
    - The approach presented in this paper seems to be ad-hoc, and it lacks rigorous experimentation to validate the contribution of building blocks of the architecture. In particular, **ablation tests** are very much necessary to clarify the contribution of different components in the proposed architecture, but no such study has been provided in the paper. Further, there is little theoretical backup and intuition behind the proposed idea.
    - The authors define Empirical Sufficiency (Definition 3.1) but do not specify the structure of the space. Specifically, it is not clear what is the underlying set, and if there is a metric associated with it? Now, depending on the structure of the space, it could give rise to many topological properties. Hence, the definition is not complete. Further, the intuitive explanation presented in the paper is not sufficient.
    - The proposed method is not rigorously compared against the existing well-known DRL techniques. The authors indeed present some experiments to highlight the improvement accomplished by their architecture w.r.t. to some hind-sight results (Table 1). However, the end-goal of solving a task (e.g. playing games) is to maximise the total reward. Hence, it is essential to compare the total reward obtained by the proposed method with that obtained by the latest DRL based algorithms, and it is missing in this paper.

**Overall impression**
The paper attacks an exciting problem but lacks rigorous experimentation and theory to validate the proposed method.

---

> ### Author Response · Authors · 2020-11-23
> **Thanks for the comments**
>
> Thanks for the comments. We would try to improve the experiments and theoretical explanation.
>
> Q1. The approach presented in this paper seems to be ad-hoc.
>
> Existing evaluation methodologies rely on the Bellman equation in-depth, however dynamic programming is not always the best way for all situations, e.g. Asymmtery sampling would cause biased estimations. We propose a new method for evaluation, which could significantly weaken uncertainty. So far we successfully prove its effectiveness in reward delay scenarios, and we would explore and exploit deeper in the future.

---

### Official Review · AnonReviewer2 · 2020-10-28

**Rating:** 5
**Confidence:** 3

**Review:**

**Paper summary**
This paper proposes a method to tackle credit assignment for delay rewards. The method is based on building a classifier that can detect states that will lead to environmental rewards in the future and assigning an additional reward to these detected states.

**Pros:**
- The overall idea is simple and intuitive.
- The visualizations in Fig 2 and 3 are clear and give good intuitions about the model.
- The results in Table 1 significantly improve over the baseline.

**Cons:**
1. The paper is not particularly easy to read. Here are some of the elements that caused me confusion and could be improved:

- Many terms introduced in the paper are not defined clearly enough (e.g. Empirical Sufficiency, Policy-based Empirical Sufficient Distribution,  Task-Specific Empirical Sufficient Distribution, Sensitive Sampling). I would suggest minimizing the introduction of new terms to the ones really required to explain the method.

- Other terms are slightly vague (e.g. pure distribution, dominant probability).

- Some terms have names that are not very intuitive (e.g. rounds, overfitting mechanism).

- Related to the previous point. In what sense is the word calibration used here? This term has a particular definition in the literature that should be refer to and justified.

In the section on Overfitting training:
- Equation (4) is not clear enough. Please define $L_E^one(\hat r_pos, r_pos) + (\hat r_neg, r_neg)$
- Are $N_{ident}$ and $N_{sufficient}$ number of rounds or a number of samples?
- What does it mean “we maximize Precision with complete $R_{negative}$ samples”?
- Is the phase 1 and 2 done in sequence or in parallel as Eq 6 suggests? In the latter case, is this the same as increasing the loss for the negative class?

In the Results section:
- Result order: The modified environment result is presented before the regular one, although in the first paragraph of the results the order is swapped.

- In Table 2, what is the difference between the red trace in top and bottom row?

2. I’m not sure the Atari levels selected for the experiments have a particularly delayed reward. With the possible exception of Bowling, the other levels usually have a short period between action and reward. Some levels that could be more interesting for testing credit assignment could be Skiing, Seaquest, Solaris or Venture (justified in e.g. Arjona-Medina et al. 2019 or Puigdomènech Badia 2020). It would be interesting to include the justification for this particular selection of levels in the paper.

3. The results are not particularly strong. From Table 2, it seems that the only significant improvement is made on FishingDerby. Is this because of the choice of levels? I would have expected a positive result in Bowling. Could the authors include a comment on why this is not the case? The results are positice in Table 1, in the modified environments, but I would like to see results in an original level with more delayed rewards (which already exist in Atari).

4. How does this method cope with intermediate (distracting) rewards in between the key action and the ultimate reward? (as in Hung et al., 2018). Is the way of selective states for the positive pools a problem to learn this type of credit assignment?

Minor:
- Tables 1 and 2 should be Figures.
- Fig3. caption “ The calibrated rewards and environmental rewards are encoded in red and yellow“  should be blue?
- “resonate with human cognition” seems a bit over-claimed to me. I would say that the visual inspections make intuitive sense.

---

> ### Author Response · Authors · 2020-11-23
> **Thanks for comments**
>
> Thanks for the contractive suggestions and pointing out the puzzling content and definition. We will carefully revise the wording.
>
> Q1. Are #N_ident# and  #N_sufficient# number of rounds or a number of samples?
>
> We introduce the definition in section 3.4. We set the number of rounds that include identified states as N_ident, and let N_sufficient denote the number of rounds that contain identified empirical sufficient states and lead to R_positive.
>
> Q2. What does it mean “we maximize Precision with complete #R_negative# samples”?
>
> The later paragraph after that sentence in section 3.4 explains the details and process. In phase two, the classifier only updates with R_negative samples. And the purpose is excluding all states insufficient to get the rewards. Once the recall of R_negative samples reaches 1, the samples recognized as R_positve could only be empirical sufficient samples to get the rewards (real R_positive).
>
> Q3. Is the phase 1 and 2 done in sequence or in parallel as Eq 6 suggests? In the latter case, is this the same as increasing the loss for the negative class?
>
> Phase 1 and 2 are done in sequence. As we train the binary classifier with R_positive samples in phase-one, and some R_positive samples and R_negative samples share the same space. The parallel training would result in endless training, where recall of R_negative might never reach 1.
> The latter case in Eq 6 does calculate the loss for the negative class.
>
> Q4. In Table 2, what is the difference between the red trace in the top and bottom row?
>
> We give the specific details in section 4.4 and description of Table 2, In the top row, the calibrated reward coefficient is set to 0.3 and the environmental reward coefficient is 1. For the bottom row, the training fully relies on calibrated rewards, where the calibrated reward coefficient is set to 1 and the environmental reward coefficient is 0.
>
> Q5. I’m not sure the Atari levels selected for the experiments have a particularly delayed reward. With the possible exception of Bowling, the other levels usually have a short period between action and reward. Some levels that could be more interesting for testing credit assignment could be Skiing, Seaquest, Solaris or Venture.
>
> We agree that the environmental feedbacks in most Atari games are not severely delayed, and this is the reason we establish hindsight reward scenarios. We previously looked at Skiing, Seaquest, etc, the delays are similar to the environments we select. Besides, we think that rewards sparsity problems have got enough attention, but rewards delay is not. There are not many eligible environments to explore the rewards delay problem, even if we do not mention this point in our paper.
>
> Q6. The results are not particularly strong. From Table 2, it seems that the only significant improvement is made on FishingDerby. Is this because of the choice of levels? I would have expected a positive result in Bowling. Could the authors include a comment on why this is not the case?
>
> There might be two reasons that the results are not significant.
> Firstly, we design ESCE as an independent network without Lstm, this is because we hope to update the ESCE network with mini-batches of unsequenced states. The adopting of mini-batches would be sensible with calculating accuracy of prediction, however, this restricts using sequenced data into the network (RNN). Proverbially, Lstm has the advantages of carrying prior information to later states. On the contrary, ESCE could only output predictions based on feature grasping on current states.
> Secondly, in actor-critic algorithms, the gradient descent of policy relies on the scalar of the estimated value. Note the two values generated (red and green in Figure 3 (a)) on key states are similar, we believe this is the reason why experiments on the original 6 games did not make much progress.
>
> Q7. How does this method cope with intermediate (distracting) rewards in between the key action and the ultimate reward?
>
> We actually briefly explored the intermediate rewards, and the results are bad as expected. It is clear that the objective for credit assignment is assigning the high credits to the crucial states, then the intermediate rewards could be considered as delayed rewards as well.

---

### Official Review · AnonReviewer1 · 2020-10-28
**Potentially good idea, but poorly presented, and with insufficient experimental support**

**Rating:** 4
**Confidence:** 4

**Review:**

The work proposes 1) classifying states according to their observed reward roll-outs and 2) using the resulting classifier for reward shaping.

This reviewer finds the work difficult to read.  For example, the term "overfitting classifier" exhibits cognitive interference with standard usage without explanation.

The use of binary classification seems most appropriate for environments with sparse rewards (both positive and negative).  Otherwise the concept of "desired environmental signals" and "undesirable environmental signals" from section 3.2 is ill-specified.  (While the reviewer can imagine a policy improvement scenario where much-better-or-worse-than-the-baseline is related to desirable/undesirable, this is not discussed.)  The experimental section is not informative with respect to the issue of defining desirable or undesirable.

A better version of this paper would justify the technique analytically.  For instance, the optimal critic would provide the true Q-values as reward.  Under what conditions does a logistic binary classifier output a probability which is close to the true Q-value (e.g., with very sparse reward rollouts)?  Or alternatively under what conditions does the classifier induce a policy gradient [equation (1)] which is a descent direction?

The experiment section at the moment is structured as "it works on these 6 atari games", but this reviewer does not find this compelling (e.g., what about the other games?  do these 6 games have an obvious desirable/undesirable reward boundary?).  However, the claims about robustness to reward delay are interesting.  This reviewer encourages the authors to try their techniques on the RWRL challenge benchmark, specifically the challenges related to reward (and observation and action) delay.

---

> ### Author Response · Authors · 2020-11-23
> **Thanks for comments and suggestions**
>
> Thanks for the advice on the wording problem and suggestions of adding new benchmarks. We do realize the idea is not sufficiently presented.
>
> Q1. The use of binary classification seems most appropriate for environments with sparse rewards (both positive and negative). (While the reviewer can imagine a policy improvement scenario where much-better-or-worse-than-the-baseline is related to desirable/undesirable, this is not discussed.)
>
> The method actually helps with the reward delay problem, but not reward sparsity problems. We introduce the desired and undesired signals in section 3.2, where desired environmental signals include positive rewards, and undesired signals include negative rewards, agent’s deaths, game endings, etc. The priority in our experiment is establishing an integrated architecture, and fairly proving the theory. In our experiments, the defining of desired / undesired signals basically is inherited from the initial purpose, which is maximizing environmental rewards.
>
> Q2. Under what conditions does a logistic binary classifier output a probability which is close to the true Q-value (e.g., with very sparse reward rollouts)? Or alternatively under what conditions does the classifier induce a policy gradient [equation (1)] which is a descent direction?
>
> This is an interesting question, we will give more details in the future. So far, we find the sufficiency extraction relies on the successes of grasping on key features and the stability of environments. Once both conditions are met, ESCE would be able to estimate a convincing probability. Also, it is worth discussing what is the true value of a state.
> In actor-critic algorithms, the gradient descent of policy relies on the scalar of the estimated value. Note the two values generated (red and green in Figure 3 (a)) on key states are similar, we believe this is the reason why experiments on the original 6 games did not make much progress.

---

### Official Review · AnonReviewer4 · 2020-10-29
**Interesting idea but poorly executed**

**Rating:** 4
**Confidence:** 3

**Review:**

The paper deals with RL problems with the reward is either very sparse or delayed, in which case the Q-values would don't get updated. Honestly, I found the write-up to be really confusing, partly because of the structure of the paper and partly because of frequent carelessness in wording. Something that would really help is to take a difficult RL problem and just use it as a running example when discussing the different concepts in the paper. For instance, how does this notion of empirical sufficiency manifest itself in an image-based example and something like grid world?

As for the results, the authors conduct experiments on 6 Atari games, which I didn't find very convincing, in particular since games like Montezuma's Revenge, which are known to be difficult because of their reward sparsity are not included. Also, there has been a fair bit of recent work in this area, so using just A3C as a baseline is hard to justify. For instance, the authors can look at https://arxiv.org/abs/1606.01868 and the many papers that followed.

In short, I think there are some interesting ideas in the paper, but both the writing and the experimental results need significant improvement.

---

> ### Author Response · Authors · 2020-11-23
> **Thanks for comments**
>
> Thank you for the approval of the idea and advice. We agree that running the methodologies on simple examples (grid world) could be a proper way of demonstration. Besides, we would also improve our write-up.
>
> Q1. The paper deals with RL problems with the reward is either very sparse or delayed, but lacks games like Montezuma's Revenge, which are known to be difficult because of their reward sparsity are not included.
>
> This paper aims at the reward delay problem, but not the reward sparsity. Intuitively, this paper introduces an evaluation mechanism (credit assignment). We construct a binary classifier, which is trained with a scheduled overfitting training process. The overfitting process could effectively exclude states’ space that may lead to more than one outcome. As a result, crucial states strongly related to environmental rewards could be accurately extracted. Thus, we believe Montezuma’s Revenge is not the proper environment for this paper.
>
> Q2. There has been a fair bit of recent work in this area, so using just A3C as a baseline is hard to justify. For instance, the authors can look at https://arxiv.org/abs/1606.01868 and the many papers that followed.
>
> The paper linked with the URL is about exploration, which focuses on the rewards sparsity problem. We introduced some recent work (credit assignment) in the related work section. We briefly mentioned our methodology is significant from previous approaches, which does not rely on the Bellman equation and the classifier distinguishes the states with certainty.

---

### Decision · Program_Chairs · 2021-01-07
**Final Decision**

**Decision:**

Reject

**Comment:**

This work tackles sparse or delayed reward problem in reinforcement learning. The key idea is to build a classifier to detect states that will lead to high rewards in the future and provide a bonus to those states. All the reviewers liked the idea but had issues with the execution of empirical results. The approach is evaluated only in a few Atari games skipping many sparse reward games and missing comparison to many exploration baselines. Furthermore, many reviewers found the writing confusing and hard to follow. The authors provided the rebuttal and addressed some of the concerns. However, upon discussion post rebuttal, the reviewers decided to maintain their score. Reviewers believe that the paper will immensely benefit with improved writing, evaluation on all atari games, and comparison to exploration baselines. Please refer to the reviews for final feedback and suggestions to strengthen the future submission.